# A One Health Perspective to Recognize *Fusarium* as Important in Clinical Practice

**DOI:** 10.3390/jof6040235

**Published:** 2020-10-20

**Authors:** Valeri Sáenz, Carlos Alvarez-Moreno, Patrice Le Pape, Silvia Restrepo, Josep Guarro, Adriana Marcela Celis Ramírez

**Affiliations:** 1Grupo de Investigación Celular y Molecular de Microorganismos Patógenos (CeMoP), Departamento de Ciencias Biológicas, Universidad de Los Andes, Bogotá 111711, Colombia; v.saenzm@uniandes.edu.co; 2Departamento de Medicina interna Universidad Nacional de Colombia, Clínica Universitaria Colombia, Colsanitas, Bogotá 111321, Colombia; caalvarezmo@unal.edu.co; 3Département de Parasitologie et Mycologie Médicale, EA1155, IICiMed, Université de Nantes, F-44000 Nantes, France; Patrice.Le-Pape@univ-nantes.fr; 4Laboratorio de Micología y Fitopatología (LAMFU), Facultad de Ingeniería, Universidad de Los Andes, Bogotá 11711, Colombia; srestrep@uniandes.edu.co; 5Facultat de Medicina I Ciéncies de la Salut, Departament de Ciéncies Médiques Básiques, Unitat Microbiología, Universitat de Rovira I Virgili, 43002 Reus, Spain; josep.guarro@urv.cat

**Keywords:** *Fusarium*, One Health

## Abstract

Any strategy that proposes solutions to health-related problems recognizes that people, animals, and the environment are interconnected. *Fusarium* is an example of this interaction because it is capable of infecting plants, animals, and humans. This review provides information on various aspects of these relations and proposes how to approach fusariosis with a One Health methodology (a multidisciplinary, and multisectoral approach that can address urgent, ongoing, or potential health threats to humans, animals, and the environment). Here, we give a framework to understand infection pathogenesis, through the epidemiological triad, and explain how the broad utilization of fungicides in agriculture may play a role in the treatment of human fusariosis. We assess how plumbing systems and hospital environments might play a role as a reservoir for animal and human infections. We explain the role of antifungal resistance mechanisms in both humans and agriculture. Our review emphasizes the importance of developing interdisciplinary research studies where aquatic animals, plants, and human disease interactions can be explored through coordination and collaborative actions.

## 1. Introduction

One Health is a concept defined as a worldwide strategy for expanding interdisciplinary collaborations and communications in all aspects of health care for humans, animals, and the environment [1]. *Fusarium* has been described as a pathogen of humans, animals, and plants, a phenomenon known as trans-kingdom pathogenicity [2,3]. In recent years, human infections by *Fusarium* have been rising worldwide, mostly involving immunocompromised hosts [3,4]. To understand these human infections, the dynamics among hosts (human/animal), pathogens, and the environment must be explored.

In humans, these fungi cause a broad spectrum of infections, including both superficial (onychomycosis and keratitis) and disseminated diseases (particularly in hematological cancer and neutropenic patients) [5]. Fungal keratitis is not only a common cause of corneal infection in developing countries, but also a significant cause of ocular morbidity and blindness [6,7]. The goal of this article is to assess the current conceptions in the One Health approach, to provide information about the host–pathogen interaction, as well to guide clinicians to consider *Fusarium* as an important human pathogen.

## 2. The *Fusarium* Genus

The taxonomy of the genus *Fusarium*, initially described by Link in 1809 as presenting banana shape conidia, has been changing in taxonomy over the years and it has become a controversial issue [8]. Currently, the genus *Fusarium* has been classified into species complexes, i.e., *F. solani* species complex (FSSC), *F. oxysporum* species complex (FOSC), *Gibberella* (Fusarium) *fujikuroi* species complex (GFSC), *F. incarnatum*–*F. equiseti* species complex (FIESC), *F. sambucinum* species complex (FSAMSC), *F. tricinctum* species complex (FTSC), *F. chlamydosporum* species complex (FCSC), and *F. dimerum* species complex (FDSC) [9,10,11]. In addition, the members of FSDC have been included in the genus *Bisifusarium* [12], and recently, it has been proposed that members of the *F. solani* species complex be moved to the genus *Neocosmospora* based on the results of phylogenetic analysis [13]. This includes the species *N. petroliphila* (*F. petroliphilum*), *N. keratoplastica* (*F. keratoplasticum*), *N. falciformis* (*F. falciforme*), and *N. solani* (*F. solani*), along with the new species *N. gamsii* (haplotype 7), *N. suttoniana* (haplotype 20), and *N. catenata* (haplotype 43) [14]. However, a recent phylogenomic analysis supports FSSC as *Fusarium* [15] and the genus name *Fusarium* for human pathogens in the FSSC [16]. Taking into account that the continuous change of names of these fungi can create some confusion, it has been suggested to refer to these taxa as *Fusarium*-like [17].

## 3. *Fusarium* in Agriculture

*Fusarium* species are broadly distributed in soil, as well as in underground and aerial plant parts, plant debris, and natural substrates [8]. Furthermore, it is found in the air and in aquatic environment, including natural sea water [18,19]. For instance, FSSC is a varied group of fungi capable of causing diseases on a wide variety of plants. It is considered the fifth plant pathogen in the list of “top ten fungal plant pathogens” [20]. On the other hand, *Fusarium* can wither ornamental plants at all production stages. In particular, *Fusarium oxysporum* f. sp. *dianthi* and its sister species, *F. redolens* f. sp. *dianthi*, can cause severe disease to carnations fields [21]. These pathogens additionally attack cereals that are important to human and animal nutrition (animal food that is found to contain mycotoxin levels above these standards would be rejected in the domestic or export marketplace) [22]. It explicitly infects certain parts of the plants, for example, grains, seedlings, heads, roots, or stems, causing various diseases, reducing product quality, and diminishing crop yields. Some species produce important mycotoxins, such as fumonisin, deoxynivalenol, and zearalenone [22].

Therefore, managing *Fusarium* outbreaks is a critical issue in agriculture. Efforts made to decrease agricultural losses by the fungal disease incorporates chemical, biological control, and fungicide use, particularly azoles compounds [23,24]. The present use rate for some triazoles, which are the biggest class of azole fungicide, is around 100 g/ha of culture [24]. The substantial use of azoles in agriculture is currently a major concern in clinical practice since it could induce antifungal resistance among diverse human fungal pathogens. Indeed, previous studies have demonstrated the presence of environmental resistance to azoles of medical importance in *Aspergillus fumigatus* isolated from flower fields related to the use of azole fungicides [25,26,27,28]. This resistance has also been reported in other environmental sources such as compost sites, public/private gardens, vineyard, and agriculture soil [29,30]. In India, a study that included isolates from patients with keratitis (agricultural workers) and environmental sources including soil, plants, gardens, parks, and agricultural fields reported that *F. falciforme* and *F. keratoplasticum* were found with high minimum inhibitory concentration (MIC) values to fluconazole, ketoconazole, and terbinafine in both clinical and environmental isolates [31]. Colombia could remain a country of concern, considering its second place worldwide as a flower exporter, including carnations, which is commonly affected by some species of *Fusarium*. About 6800 hectares are destined for the cultivation of flowers for export, from which 75% are located in the savannah of Bogotá [32]. Based on the Colombian Agricultural Institute report, fungicides acting by inhibition of ergosterol synthesis are approved to be used in the flower industry [25,33]. This data foregrounds the necessity to study *Fusarium* in crops, flower fields, and clinical settings.

### Fusarium in Hospital Environment

Water distribution systems (drains, faucet aerators, shower heads) in hospitals have been identified as potential reservoirs for species of *Fusarium*. These distribution systems are thought to be responsible for nosocomial infections [34]. Most of the pathogenic species of these fungi have been found in environmental samples, including plumbing systems of hospitals [35]. During hospital outbreaks where immunosuppressed hosts are affected, using a genotyping approach has shown a relationship between hospital water and patient isolates. This suggest that shower is most likely to be the mechanism of aerial dispersal of the conidia in air: it could be responsible for the transmission to the host [34,36]. Furthermore, airborne conidia may also represent a relevant source of infection as occurred in the case of poorly sealed chase openings permitting inadequate air exchange and exhausts [37]. A recent study evidenced a genetic relationship between *Fusarium* species isolated from indoor hospital air with the ones recovered from blood cultures of hematologic patients, which suggest that the air may be a potential source for fusariosis [4]. Airborne fusariosis is considered to be acquired by the inhalation of airborne conidia, as indicated by the occurrence of sinusitis and/or pneumonia in the absence of dissemination [38].

## 4. *Fusarium* and Animal Disease

*Fusarium* species represents a common cause of opportunistic infections in aquatic animals, such as seahorses, turtles, dolphins, pinnipeds, and colonization of eggs. *Fusarium* infection is considered as the primary driver of declining turtle populations around the world [39]. The sea turtle egg fusariosis (STEF) is an emerging disease and is responsible for egg mortality in sea turtles around the world, where most of the cases are related to *F. keratoplasticum* and *F. falciforme* [40,41,42,43,44,45]. During the embryonic development, the eggs spend an extensive amount of time buried under the sand in a sticky and warm consistent temperature, and it has been suggested that these conditions support the development of soil-borne fungi such as *Fusarium* [41,45]. Clinical manifestations suggest that STEF possibly include the presence of atypically colored areas (e.g., yellow, blue, gray, red) on the eggshell, with more severe infections showing gray hyphal mats on the outside and the inside of the eggs, as well as in the embryos [45]. The clinical signs of infection caused by *Fusarium* in aquatic animals shift from a superficial invasion of the skin to pulmonary or systemic infections affecting the lungs, the liver, the heart, and the cartilages [39]. Moreover, an infection caused by *F. oxysporum* in captive-reared fingerlings of golden mahseer has been recently informed in India [46].

Some of the principal hypotheses on the STEF etiology and epidemiology lie in the relationship between beach environments and human environmental sources. *F. keratoplasticum* and *F. falciforme* are probably microbiota of beach sand, and there is a relationship between human construction plumbing systems and STEF disease. These pathogens are brought into the sand beach by overflow from plumbing human wastes [41].

Other cases related to animal pathologies are equine keratitis infections [47], in addition to invasive sinusitis and facial mycetoma in dogs [48]. Diversely, raised concentrations of fumonisins in animal feed have been accounted to cause sicknesses, such as Equine Leukoencephalomalacia (ELEM) in horses, and porcine pulmonary edema (PPE) or liver injury in pigs [22]. Trichothecenes mycotoxines produced by *Fusarium* develop an important role in animal health due to their potential to be topically absorbed, and their metabolites affect the gastrointestinal tract, skin, kidney, liver, and immune system [49].

## 5. *Fusarium* in Human Diseases

In severely immunocompromised patients, fusariosis is the second most common mold infection in humans, right after aspergilosis [9]. These fungi cause superficial (such as onychomycosis and keratitis), locally invasive, and disseminated disease.

### 5.1. Onychomycosis

Onychomycosis is one of the most widely recognized finger and toenail infections with an overall prevalence of 5.5% [50]. This pathology can affect the physical, functional, psychosocial, and emotional state of the patient [51]. Even though it is not a life-threatening condition, numerous significant anatomical functions of the nail may be affected, with difficulty in walking, embarrassment, and work-related challenges being the most commonly reported issues [51]. Recognized risk factors for onychomycosis are trauma, ageing, obesity, diabetes, participation in fitness activities, immunosuppression (HIV(human immunodeficiency virus), drug-induced), malignancies, sedentarism, and occlusive footwear [50,51,52]. People who have pedicure treatment are less likely to acquire onychomycosis [53].

Dermatophytes, mainly *Trichophyton rubrum*, are responsible for most fungal nail infections and about 30% to 40% of onychomycosis cases are caused by nondermatophyte molds (NDMs) and yeasts [50]. In South America, studies suggest that *Fusarium* may be the most common NDM [53,54]. Species identification has a crucial role in this disease, for example, *F. keratoplasticum* and *F. falciforme* are the most frequent species isolated in Colombia, and some of these isolates exhibited lower azole in vitro activity [53].

Fungal production of proteases that degrade keratin may facilitate invasion [53]. In addition, histological studies have revealed the capacity of *F. oxysporum* to invade human nails, including the firm attachment to the nail plate and the dissemination to deep layers, causing disorganization of nail structure [55]. Also, the formation of fungal biofilms is a contributor to persistent infection, which offers advantages such as antifungal resistance, protection against host defenses, increased virulence, communication, metabolism cooperation, and differential gene expression [56,57,58]. By viewing an infected nail through a scanning electron microscope (SEM) it can be seen that this fungus is able to form a biofilm, by penetrating unassisted nail layers to cause onychomycosis. It can also be seen that the ventral surface of the human nail is more vulnerable to infection than the dorsal surface [56].

### 5.2. Keratitis

Corneal disease is one of the leading causes of blindness worldwide. In 2001, trauma and corneal ulceration were reported as principal causes of unilateral blindness and the global estimate varies from 1.5 to 2 million cases per year [6]. Current epidemiological information proposes that microbial keratitis might be epidemic in South, South-East, and East Asia, and may exceed 2 million cases per year worldwide [59]. Furthermore, it has been demonstrated that fungal keratitis contrasted to bacterial keratitis can be progressively destructive. On the one hand, a retrospective analysis not only found that fungal keratitis was more likely to perforate the cornea than bacterial keratitis (OR (Odds ratio) = 5.86, 95% CI (confidence interval), 1.35–20.66), but also that it leads to an irreversible change [60]. Likewise, around 15–27% of patients with fungal keratitis require surgical intervention, such as corneal transplantation, removal of ocular contents, and enucleation, as a result of a failed pharmacological treatment [61].

The requirement for prolonged and intensive treatment, resulting in negative ocular outcomes (loss of vision and/or loss of the eyeball), indicate that both the economic and medical implications are substantial [62]. In addition, a correlation between gross national income (GNI) and the etiology of microbial keratitis has been shown. Fungal keratitis is associated with countries with low GNI [63]. Moreover, there are numerous cases of fungal keratitis related to the utilization of contact lenses. Between 2004 and 2006, an outbreak of *Fusarium* fungal keratitis occurred in contact lens users worldwide, owing to a decrease in the disinfection capacity of contact lens solutions [64]. Clinical signs of fungal keratitis include a sudden onset of pain along with photophobia, discharge with a reduced vision, and opacity on the surface of the cornea [65].

The common fungal causative agents are *Fusarium* spp., *Aspergillus flavus* and *A. fumigatus*, and *Candida albicans* (which is less common in tropical climates) [66]. *Fusarium* keratitis has increased over the last forty years and it is estimated that around half of all the cases of microbial keratitis in tropical countries are caused by this genus, probably on account of an increasing use of topical steroids and antibacterial agents, as well as a rise in surgical procedures, contact lens use, ocular trauma, chronic ocular surface diseases, and immunocompromised patients [67]. In Tunisia, fungal keratitis represents 83% of the cases, with *F. solani* being the most prevalent species (66%) [68]. In Brazil, 25% of fungal keratitis is caused by *Fusarium* [69], and in Mexico, *F. solani* was found in 37% of the patients [70]. Recent studies conducted in south India have shown that *F. keratoplasticum* and *F. falciforme* were the most prevalent species isolated from keratomycoses and environmental settings; in fact, agricultural workers in India often become infected after a corneal injury caused by plant or soil material [31].

The interaction of pathogenic fungi with host cells is the main factor in the pathogenesis of mycotic keratitis. The human central corneal temperature (32.6 ± 0.70 °C) is suitable for the development of *Fusarium* [71]. Adherence of microorganisms to host cells through an assortment of adhesins is essential for the initiation of the infection [65]. Consequently, *Fusarium* keratitis can invade the cornea and the anterior chamber of the eye. Here, in the pupillary area, it forms a lens-iris-fungal mass which affects the normal drainage of the aqueous humor and causes an increase in the intraocular pressure, leading to fungal malignant glaucoma [65,72]. Also, *Fusarium* mycotoxins can suppress immunity and break down tissues. Certain cytosolic proteins and peptide toxins can destroy corneal epithelial cells [73]. Proteases play an important role in fungal keratitis because they can cause corneal ulcers [65,74]. As described in onychomycosis, the formation of biofilm is another factor that contributes to the pathogenesis of keratitis as well as to antifungal resistance [75,76]. Biofilm proteomics studies in *F. falciforme* have identified several proteins whose levels changed during the biofilm formation phases, as well as the enzymes involved in the glycolysis/gluconeogenesis and pentose pathways. Some of the proteins involved could promote angiogenesis, adhesion/invasion, and immunomodulation [76].

### 5.3. Invasive Disease

Invasive fusariosis affects most patients with prolonged and profound neutropenia and/or severe T-cell immunodeficiency, acute leukemia, and hematopoietic cell transplant (HCT) recipients. Besides, it is not only the most frequent clinical form of fusariosis but also the most common challenging in immunocompromised patients, accounting for approximately 70% of all cases of fusariosis in this population [38]. A retrospective analysis of 233 cases (92% of them being patients with hematologic diseases) reported that the outcome is usually poor, with a 90-day probability of survival of 43% of the patients [77].

The typical clinical onset consists of a patient with prolonged (>10 days) and profound (<100 cells/mm^3^) neutropenia who is persistently febrile and develops disseminated and characteristic skin lesions (papular or nodular erythematous lesions), with a positive blood culture [38,78]. *Fusarium solani* is the most common species involved in fusariosis (50% of cases), followed by *F. oxysporum* (20%), and *F. verticillioidis* and *F. moniliforme* (10% each) [4].

In relation to pathogenesis, animal models of fusariosis showed that mortality was correlated with inoculum size [38]. In nonneutropenic mice, the disease was described by necrotizing abscesses with hyphae, hemorrhage, and neutrophil and macrophage infiltration [79]. Paradoxically, neutropenic mice did not exhibit an inflammatory cellular reaction and had a significantly higher fungal burden [79]. A murine model of intratracheal inoculation of *F. solani* was recently used to investigate its spread to different organs in immunocompetent animals within 24 h after inoculation. Results showed that a 1 × 10^8^ conidia/animal inoculum followed a 100% death rate of immunocompetent mice in 24 h [80].

## 6. Fusariosis Treatment

Before reviewing any human fusariosis treatment, we must discuss a frequently forgotten issue in clinical practice: the role of the environment and fungicides in the patient response to antifungal drugs. Fungicides are chemical agents utilized for control and treatment of fungal infections in plants. They exhibit a variety of mechanisms of action, such as effects on respiration, signal transduction, mitosis cell division, membrane, and cell wall [81]. Also, azole fungicides, which include tebuconazole, propiconazole, and epoxiconazole, also called demethylation inhibitor (DMI), are the most widespread treatment in agriculture due to their low cost and broad spectrum [82]. For example, tebuconazole is generally used to control FHB (*Fusarium* head blight) [83]. Tebuconazole demonstrated various effects on *Fusarium culmorum* (a common pathogen of cereals), including morphological changes at the ultrastructural level such as considerable thickening of the hyphal cell walls, excessive septation, the formation of the incomplete septa, extensive vacuolization, accumulation of lipid bodies, and progressing necrosis or degeneration of the hyphal cytoplasm [84]. Morever, *F. culmorum* is capable of adapting to triazole pressure by overexpressing a drug resistance transporter [85].

As referenced before, the overuse of fungicides in crops and flower fields becomes imperative for the identification of *Fusarium* to the species level (some species have higher MIC values than others), not only from an epidemiological viewpoint but also for choosing the appropriate antifungal treatment [86].

### 6.1. Localized Infection

There are currently no available antifungal recommendations in accordance with *Fusarium* isolation. Treatment with nail lacquers and systemic treatment is usually used; unfortunately, *Fusarium* onychomycosis and keratitis are difficult to eradicate. Onychomycosis systemic treatment with itraconazole or terbinafine is usually effective, but relapses are very common [87]. Some *Fusarium* strains isolated from nail samples have also demonstrated in vitro susceptibility to amphotericin B (which binds to ergosterol in the cell membrane) [53]. Additionally, treating fungal keratitis represents a challenge because of the limited and variable susceptibility of *Fusarium* to antifungal agents, the poor tissue penetration of topical antifungal agents, resulting in low drug bioavailability, and the absence of routine determination of antifungal susceptibility [88]. First-line therapy for *Fusarium* keratitis includes a topical antifungal agent either alone or in combination with systemic antifungal medication. Natamycin (which inhibits fungal growth by binding to sterols) has been the traditional drug of choice for topical treatment; however, amphotericin B drops (1.5 mg/mL) and voriconazole have also been used [89]. A randomized trial comparing topical 5% natamycin with topical voriconazole 1% for the treatment of fungal keratitis (24.6% of which were caused by *Fusarium*), suggested that natamycin may be more effective in healing corneal ulcers and improving visual acuity [90].

### 6.2. Invasive Infection

There is a variable susceptibility of *Fusarium* spp. to antifungal agents. The empirical treatment for invasive fusariosis infections is either voriconazole (VRC) (which inhibits the ergosterol production by binding and inhibiting the lanosterol-14alpha-demethylase), or liposomal amphotericin B (L-AMB), surgical debridement (if conceivable), and posaconazole (which inhibits the ergosterol production by binding and inhibiting the lanosterol-14alpha-demethylase) for salvage therapy [91]. If possible, neutropenia recovery and surgical debridement could be disease management tools. Information displays a 90-day survival rate of 42% in patients treated with voriconazole and showed that combined therapy does not work better than voriconazole alone [92]. In patients with acute leukemia, L-AMB or VCR are preferred. The ending point of invasive infection greatly depends on persistent neutropenia and or corticosteroid-induced immunosuppression [93]. In vitro synergism between antifungals and antimicrobials or non-antifungal agents have been studied, and percentages of synergism were as high as 80% for amiodarone (AMD) + VRC, of 75% for moxifloxacin and AMB, and of 65% for AMD + AMB [94].

## 7. Antifungal Resistance Mechanisms

In general, the members of *Fusarium* have shown primary or secondary resistance to practically all currently used antifungals, such as azoles, echinocandins, and polyenes [95]. An organism that is resistant to a drug prior to exposure is described as having primary or intrinsic resistance. Secondary resistance develops in response to exposure to an antimicrobial agent [96]. Both of these mechanisms have been reported in these fungi, although the molecular mechanisms of intrinsic resistance have not been described yet [11]. Secondary resistance to azoles has been demonstrated in *A. fumigatus* and is usually dependent on an altered expression of *CYP51*, the gene encoding sterol 14α-demethylase [97,98]. In the *Fusarium* genus, there are three paralogous *CYP51* genes which have been depicted and assigned as *CYP51A*, *CYP51B*, and *CYP51C.* In agriculture samples, the overexpression of *CYP51A* in *F. graminearum* in the presence of tebuconazole has already been described [99]. More recent evidence shows that *F. keratoplasticum CYP51A* mRNA levels are ∼6500-fold upregulated in response to azole antifungals to compensate for the loss of *CYP51B* function due to azole inhibition. A strong association of voriconazole resistance with a 23 bp CYP51A promoter deletion in *F. keratoplasticum* isolates was also demonstrated [100]. Lineage-specific (LS) chromosomes have been described in several plant pathogenic filamentous fungi. A recent study discovered the presence of LS in two *F. oxysporum* clinical strains with an important role in niche adaptation and resistance, for example, a doubling of genes coding for ergosterol synthesis and, in addition, more than 70 copies of genes coding for various efflux pumps [101].

Furthermore, it was reported recently that several genes of *F. oxysporum* and *F. solani* related to mechanisms of antifungal resistance such as ergosterol synthesis pathways, drug efflux pumps, response to oxidative stress, and cell wall biosynthesis, were differentially regulated upon the treatment with amphotericin B (AMB) and posaconazole (PSC) [102].

## 8. One Health Perspective

The emphasis of the One Health approach may explain common intersecting points, such as STEF disease, and plumbing human contaminations, hospital water, and air distribution systems are reservoirs for the fungi. Other risks of human infection come from environmental sources, for example, keratitis may develop following the traumatic inoculation with *Fusarium*-contaminated soil or plant material (Figure 1).

Furthermore, the extensive use of azoles fungicides in agriculture leading to a risk of antifungal resistance in humans is a significant issue of concern. Azole fungicides utilized in agriculture share the same mechanism of action (inhibition of lanosterol-14α-demethylase) and similar molecular structures (pharmcophores) as medical triazole drugs [103]. Furthermore, a recent rate of emergence of fungicide-resistance pathogens has been reported [104]. Overall, all these findings challenge clinicians and researchers to understand every part of the puzzle. We must continue efforts to think outside the box, and work together with all actors of One Health initiatives. One Health must integrate fungal diseases into health systems [105] and improve interdisciplinarity studies that include the patient, the environment, and the relationship among all the elements that affect fusariosis in agriculture, and human and animal diseases.

## 9. Conclusions

Data from fusariosis in animals and plants are vital to understanding human infections. Fungicide controls fungal diseases but many fungicides share target activity with antifungals. This explains why the high use of fungicides in agriculture is risky for developing resistance to antifungals in clinical practice. To recognize fusariosis disease correctly in humans, animals, and plants, epidemiological information and research should be done across all segments. Government, agriculture, clinical, veterinarian, and plant authorities must initiate joint actions in order to respond to the One Health approach in fusariosis.

## Figures and Tables

**Figure 1 jof-06-00235-f001:**
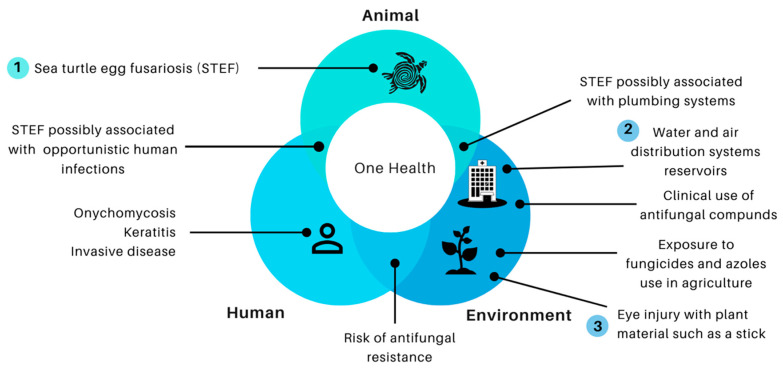
Animal, environment, and human interaction: a One Health perspective.

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
