# Peer review of "A One Health Perspective to Recognize Fusarium as Important in Clinical Practice"

_jof, 2020, doi:10.3390/jof6040235_

Round 1

Reviewer 1 Report

In principle the "One Health" concept is interesting and furthermore Fusarium is a good example for this discussion. But the authors have failed to convince the reader. They largely discuss the human diseases caused by this heterogenous group of fungi. But they do not show the connection between these findings and one health. In contrast, they cite examples of Fusarium spp. which are exclusively plant pathogenic and do not play any role neither in human and veterinary medicine  nor in the environment. The role of mycotoxins produced by plant pathogenic fungi dependant on environmental conditions like temperature, humidity etc. and concentrated in food stored and stable in the environment for long time on animal and human health is only marginally respected, although this aspect is quite characteristic for "one health".

Thus, this paper gather many information about Fusarium but the connection with the concept of "one health" is only partially successful. A much better title would be: Fusarium spp.are versatile organisms.

Title: Since Neocosmospora belongs more or less to Fusarium spp. this name should be omitted in title (and even later), since for most readers this fungal name will be confusing

Line 17-18: In this first sentence the term "one health" should be explained to the reader, because not all are familiar with. 

Line 21-23: akward: proposition: The concept of "one health" implies that the microbiological aspects of human and veterinary medicine should be evaluated at the same time with plant protection and environmental habitats.

Line 33; 43; 62; 92; 95;108;131: omit Neocosmospora

Line 64: "found in aquatic habitants" habitats?

Line 71-74: By the way these examples vote against "one helath", because these strains obviously have no human pathogenic capacities which endanger human and animal health

Line 78: The half-time of azoles in earth is about 365 days!

Line 86: By the way a recent publication gives an explanation, since in the genome of Fusarium multicopies of resistance mechanisms are present:
Zhang, Y.; Yang, H.; Turra, D.; Zhou, S.; Ayhan, D.A.; DeIuli, G.A.; Guo, L.; Broz, K.; Wiederhold, N.;Coleman, J.J.; et al. The genome of opportunistic fungal pathogen Fusarium oxysporum carries a unique set
of lineage-specific chromosomes. Commun. Biol. 2020, 3, 50. 

Line 107: In principal Fusarium spp. are introduced in to the hospital area and wards by the human microbiome. The patients carry Fusarium spp. in the gut!

Line 113: Again: an argument against "one health" since these fungi are non pathogenic for humans

Line 130: The aspects of infection and intoxication should be clearly separated. Indeed the production of toxins (especially of trichothecenes) should be considered in this context much more since it is an example that Fusarium in agriculture has a role for human health.

Line 147: This list is not complete. For example out of the keratinophilic species only Scopulariopsis but not Chrysosporium and Wallemia are noted. I suggest to omit the list, because this info is not essentila in the context of the paper.

Line 192-207: What does this paragraph contribute to the topic "one health"?

Line 221-227: This paragraph describes animal disease under the heading of human disease, The wrong place

Section 6: In the genome of F. oxysporum, one can find a doubling of genes coding for ergosterol synthesis and, in addition, more than 70 copies of genes coding for various efflux pumps (Zhang Y et al., 2020)

Section 6, L 242-245: This aspect of misuse of azoles in agriculture is already discussed above. Furthermore, this discussion is unidirectional: the most active drug for the treatment of human fusariosis is amphotericin B. 

Section 6.1: What is the relationship of this paragrapgh to "One health"?

Line 270: What is the role of terbinafine?

Line 280: See Zhang Y et al., 2020)

Section 7: The relevance of this paragraph for "one health"?

Author Response

We really appreciate your comments and suggestions, they are a very good feedback to our work and the corrections have been already made in the document. You can find the corrections in the manuscript since we are using track changes

Comment: In principle the "One Health" concept is interesting and furthermore Fusarium is a good example for this discussion. But the authors have failed to convince the reader. They largely discuss the human diseases caused by this heterogenous group of fungi. But they do not show the connection between these findings and one health. In contrast, they cite examples of Fusarium spp. which are exclusively plant pathogenic and do not play any role neither in human and veterinary medicine nor in the environment. The role of mycotoxins produced by plant pathogenic fungi dependent on environmental conditions like temperature, humidity etc. and concentrated in food stored and stable in the environment for long time on animal and human health is only marginally respected, although this aspect is quite characteristic for "one health".

Answer: Thank you for pointing out that our manuscript could be understood in this way. Our main purpose is to connect the environment, animals, and human disease in one health approach. In particular, the number of reports of Fusarium species that were previously considered to be exclusive plant pathogens but are now implicated in superficial and systemic infection in humans and animals is increasing (Zhang, et.al 2006). For example, members of the Fusarium solani species complex (FSSC) group of fungi capable of causing diseases on a wide variety of plants are also the most prevalent species isolated from human mycotic keratitis (Homa, M. et.al 2018) and one of the principal causes of the sea turtle egg fusariosis (Cafarchia et al., 2020 and Smyth et al., 2019). A recent paper from Meza-Menchaca (2020) investigated whether Fusarium species isolated from keratomycosis human patients, which are normally pathogens of plants, conserve their infective capacity to re-infect plants and other human tissue and demonstrated that the same fungal pathogen was able to produce both diseases.

Comment: Thus, this paper gather many information about Fusarium but the connection with the concept of "one health" is only partially successful. A much better title would be: Fusarium spp. are versatile organisms.

Answer: Thank you for the suggestion, we decided to focus this paper in the One Health approach because we agree that is necessary a collaborative, holistic surveillance of the environment, animals, and humans in fusariosis. The American Academy of Microbiology held a colloquium in 2019 titled “One Health: Fungal Pathogens of Humans, Animals, and Plants” (Konopka JB, Casadevall A, Taylor JW, Heitman J, Cowen L. 2019.) The Global Action Fund for Fungal Infections (GAFFI) and the World health organization encourage joint efforts in the diagnosis and treatment of animal and human diseases, including the responsible use of antifungals; support cross-kingdom disease surveillance that can aid in the early recognition of emerging infectious diseases. Based on this our main propose with this manuscript is to recognize Fusarium as an important pathogen that has implications in the one health triad.

Comment Title: Since Neocosmospora belongs more or less to Fusarium spp. this name should be omitted in title (and even later), since for most readers this fungal name will be confusing

Answer: Thank you for these observations we have remove Neocosmospora from the title. Please see track changes in line 2.

Comment: Line 17-18: In this first sentence the term "one health" should be explained to the reader, because not all are familiar with. 

Answer: As suggested, we added the sentence “a multidisciplinary, and multisectoral approach that can address urgent, ongoing, or potential health threats at the human-animal-environment”. (Please see lines 21-22)

Comment: Line 21-23: akward: proposition: The concept of "one health" implies that the microbiological aspects of human and veterinary medicine should be evaluated at the same time with plant protection and environmental habitats.

Answer: Thank you for pointing this out. Although we agree that this is an important consideration, it cannot be analyzed at the same time due to the limitations of studies that address a one health approach, investigations in this field, unfortunately, are separated. One of the main reasons for writing this manuscript and reflections that we mention in the conclusion.

Comment: Line 33; 43; 62; 92; 95;108;131: omit Neocosmospora

Answer: We agree with the reviewer’s assessment. Accordingly, throughout the manuscript, we have removed the term Neocosmospora from lines: 30, 35, 37, 51, 69, 98, 105, 107.

Comment: Line 64: "found in aquatic habitants" habitats?

Answer: Thanks for pointing this out. Due to the changes, this line now corresponds to line 71. The word habitats and change to environment have been removed. Please see line 71.  

Comment: Line 71-74: By the way these examples vote against "one health", because these strains obviously have no human pathogenic capacities which endanger human and animal health

Answer: We think this is a good suggestion. We have added a sentence to improve the impact in animal health “animal feed that is found to contain mycotoxin levels above these standards would be rejected in the domestic or export marketplace”. Due to the changes, this line now corresponds to line 78

Comment: Line 78: The half-time of azoles in earth is about 365 days!

Answer: Thank you! These data underline the need to determine the prevalence of azole resistance in Fusarium in clinical and environmental settings. Due to the changes, this line now corresponds to line 86

Comment: Line 86: By the way a recent publication gives an explanation, since in the genome of Fusarium multicopies of resistance mechanisms are present:  Zhang, Y.; Yang, H.; Turra, D.; Zhou, S.; Ayhan, D.A.; DeIuli, G.A.; Guo, L.; Broz, K.; Wiederhold, N.;Coleman, J.J.; et al. The genome of opportunistic fungal pathogen Fusarium oxysporum carries a unique set
of lineage-specific chromosomes. Commun. Biol. 2020, 3, 50. 

Answer: We would like to thank the reviewer for pointing out this very interesting work, we have included this information in section 7. “Lineagespecific (LS) chromosomes has been described in a number of plant pathogenic filamentous fungi a recent study discovered the presence of LS in two F. oxysporum clinical strains with important role in niche adaptation and resistance [99]. Due to the changes, this line now corresponds to line 318-321

Comment: Line 107: In principal Fusarium spp. are introduced in to the hospital area and wards by the human microbiome. The patients carry Fusarium spp. in the gut!

Answer: Thank you! A mutual interaction between mycobiome and microbiome in the gut can be assumed as mention by Hof 2020 (The Medical Relevance of Fusarium spp Journal of Fungi 2020 6(3):117) But the relevance is still largely unknown. We would appreciate if the reviewer could provide more information in this point.

Comment: Line 113: Again: an argument against "one health" since these fungi are non pathogenic for humans

Answer: As mentioned in the manuscript the sea turtle egg fusariosis (STEF) is a recently emerging disease and is responsible for egg mortality in sea turtles around the world; most of the cases related to    F. keratoplasticum and F. falciforme, these species are the most prevalent isolated from human mycotic keratitis and disseminated disease as we discuss in section 5.2 (Due to the changes, this line now corresponds to line 127)

Comment: Line 130: The aspects of infection and intoxication should be clearly separated. Indeed the production of toxins (especially of trichothecenes) should be considered in this context much more since it is an example that Fusarium in agriculture has a role for human health.

Answer: We have added the suggested content to the manuscript on line 144-147

Comment: Line 147: This list is not complete. For example, out of the keratinophilic species only Scopulariopsis but not Chrysosporium and Wallemia are noted. I suggest to omit the list, because this info is not essentila in the context of the paper.

Answer: We thank the reviewer for this observation. As suggested, we omit the list. Please see track changes. Due to the changes, this line now corresponds to line 174

Comment: Line 192-207: What does this paragraph contribute to the topic "one health"?

Answer: While we understand the reviewer's concern regarding the contribution to the topic “one health” we would like to point out that the goal of this revision is to provide an assessment of the current conceptions in one health approach and host-pathogen interaction. Line 192-207 (due to the changes, this line now is line 227-242) correspond to the pathogenesis we explain in each section of Fusarium in human diseases.

Comment: Line 221-227: This paragraph describes animal disease under the heading of human disease, The wrong place

Answer: We thank the reviewer for this observation. We would like to point out that in this case, we are giving examples of animal models related to invasive disease pathogenesis. (due to the changes, this line now is line 257-263)

Comment: Section 6: In the genome of F. oxysporum, one can find a doubling of genes coding for ergosterol synthesis and, in addition, more than 70 copies of genes coding for various efflux pumps (Zhang Y et al., 2020)

Answer: We would like to thank the reviewer for pointing out this very interesting work, we have included this information in section 7 reference 99.

Comment: Section 6, L 242-245: This aspect of misuse of azoles in agriculture is already discussed above. Furthermore, this discussion is unidirectional: the most active drug for the treatment of human fusariosis is amphotericin B. 

Answer: We thank the reviewer for this observation. We would like to point out that due to the spectrum of human diseases caused by Fusarium (superficial to disseminated diseases) amphotericin B is not always the most active drug and the high exposure of environmental fungi to fungicides may increase the risk of resistance development.

Comment: Section 6.1: What is the relationship of this paragraph to "One health"?

Answer: While we understand the reviewer's concern regarding the contribution to the topic “one health” we would like to point out that the main goal of section 6.1 is to provide information about treatment. Fusarium is difficult to treat in clinical practice due to antifungal resistance and few antifungals drugs available. One of the aims of one health is also to include global initiative for revised regulations for the use of azoles in therapeutic treatments and to develop new drug targets.  

Comment: Line 270: What is the role of terbinafine?

Answer: Thank you for pointing this out. Terbinafine has not role in invasive infection.

Comment: Line 280: See Zhang Y et al., 2020)

Answer: We would like to thank the reviewer for pointing out this very interesting work, we have included this information in section 7. (due to the changes, this line now is line 322)

Comment: Section 7: The relevance of this paragraph for "one health"?

Answer: While we understand the reviewer's concern regarding the contribution to the topic “one health” we would like to point out that the main goal of section is to inform the main mechanism of antifungal resistance, by understanding how Fusarium develops resistance, effective measures can be designed and implemented preventing resistance development.

Reviewer 2 Report

The manuscript by Sáenz et al. describes the increased concern in the impact that Fusarium and Neocosmospora have on animal, plant, and human infections. It is a short and solid review manuscript that I enjoyed to read. I thank the authors for this. The manuscript contains really minor typos and grammar mistakes that could be addressed after acceptance for publication.

Other points for the authors' consideration:

-Lines 17 and 32 should include plants too.

-I think that the title of section 2 should be changed to "The Fusarium genus". In addition, the content of this section could be complemented with a figure dealing with all these taxonomical changes.

-Line 64, please specify whether is fresh or seawater.

Line 120, pulmonary would restrict the disease to animals with a respiratory apparatus like ours, what about those with gills?

Line 132, I think the authors should double-check this fact, I do not think that Fusarium or Aspergillus are more frequent than dermatophytes.

Line 141, what about sedentarism?

Author Response

We really appreciate your comments and suggestions, they are a very good feedback to our work and the corrections have been already made in the document. You can find the corrections in the manuscript since we are using track changes

The manuscript by Sáenz et al. describes the increased concern in the impact that Fusarium and Neocosmospora have on animal, plant, and human infections. It is a short and solid review manuscript that I enjoyed to read. I thank the authors for this. The manuscript contains really minor typos and grammar mistakes that could be addressed after acceptance for publication.

Answer: Thank you for your feedback and careful review.

Comment: Lines 17 and 32 should include plants too.

Answer: Thank you for the suggestion. In the one health approach plants are included in environment.

Comment: I think that the title of section 2 should be changed to "The Fusarium genus". In addition, the content of this section could be complemented with a figure dealing with all these taxonomical changes.

Answer: Thank you for these observations we have changed to “The Fusarium genus” please see line 52. We consider that Geiser 2020 paper under review and ODonnel 2020 article in press make an excellent contribution to taxonomy in our case this section is just informative of taxonomy changes but not the main goal of this review.

Comment: Line 64, please specify whether is fresh or seawater.

Answer: Thank you for pointing this out. We specify that is seawater in line 71 (due to the changes, this line now is line 71)

Comment: Line 120, pulmonary would restrict the disease to animals with a respiratory apparatus like ours, what about those with gills?

Answer: We think this is an excellent suggestion. (due to the changes, this line now is line 136). We have included the sentence “And recently it has been informed about and infection due to F. oxysporum in captivereared fingerlings of golden mahseer” (line 136-138)

Comment: Line 132, I think the authors should double-check this fact, I do not think that Fusarium or Aspergillus are more frequent than dermatophytes.

Thank you for pointing this out. (due to the changes, this line now is line 164). To clarify this point we have added the sentence “In severely immunocompromised patients”.

Comment:  Line 141, what about sedentarism?

Answer: Thank you for pointing this out. (due to the changes, this line now is line 174). We have included the word sedentarism

Reviewer 3 Report

The article entitled "A one health perspective to recognize Fusarium and Neocosmospora as important in clinical practice" by Valeri Sáenz and coworkers is an interesting attempt to advise regarding the use of fungicides in Agriculture or other field and their impact in human, crop, and animals health.

The article is, in general, well organized and well written. However, in addition to its objective, it introduces a question that is currently under an intense discussion in the Fusarium community, as is the use of the name Neocosmospora for some formerly named Fusarium species. Further, in the manuscript, it is not clear what name the authors are using, and this introduces a point of unnecessary distraction for the reader.

I would suggest the use of only the name of Fusarium for all examples used, waiting to see in the near future what is the conclusion reached by the general Fusarium community. As the authors indicated, there are currently several important articles accepted or being evaluated that go in-depth with this question.

I found some other points that could be improved, which are listed below.

Lines 84-86, from In India,.. to terbinafine.
This sentence has no sense in this form, please improve it.

Line 92, and all the introduction section
The inclusion of some Fusarium (former) species in the genus Neocosmospora is currently under an intense discussion. However, in the article, authors use the new names even when they indicated at the beginning of the article that will use the name Fusaria-like to for those species under study. Please, authors have to try to be consistent with the nomenclature used in the present article, in which, the objective is not to solve the discrepancies in those species names.

Line 93, head of section 3.1/ Line 108, head of section 4- and following sections.
See the previous comment.

Section 3.1.
Similarly to what it was explained for airborne Fusariosis, authors could go further in describing the more frequent fungal infections caused by water transmitted Fusarium conidia.

Line 109.
At this point, authors use the term Fusaria-like. Please, the use of this term only in specific points of the article introduces a point of distraction for the reader, please be consistent in the entire article regarding the nomenclature used. This article is interesting itself, authors do not need, I insist, to go further in the nomenclature controversy. Take this into account for the entire article.

Lines 146-147.
I suggest the authors to list the names of the NDMs organisms in order regarding their higher-to-lower incidence. Probably the first in this list should be Fusarium spp.?

Line 148.
An additional term is used at this point: fusarium-like. Is this the same as Fusaria-like?. Please revise.

Paragraph starting in line 151.
The mechanism described for Fusarium infection in onychomycosis is not exclusive of this fungus, it is the usual way to start this infection for all known onychomycosis-causing agents. Please, improve this paragraph.

Lines 201-202. Sentence starting with “The PacC/Rim…..ocular surface” is irrelevant in this manuscript´s context. Maybe, authors could move this sentence to the end of this section.

Lines 233-235. From “Azole fungicides….” to “broad-spectrum”.
This sentence is out of context.

Section 6.
General comment to this section.
Authors do not really talk about the fusariosis treatment, this section only includes a summary of only a few antifungal compounds, but it is not an exhaustive review of the diverse possible antifungal treatments. Please, improve or change the head of this section.

Section 6.1.
Please, include some detail regarding the characteristics of amphotericin B and natamycin, i.e. type, targets… Even when both are common antifungal agents, mainly amphotericin B, it will be easier to follow for non-specialized readers.

Section 6.2.
See previous comment, to be applied for Posaconazole at this point. Please revise further examples of antifungal agents to improve their description in the base of these comments.

Author Response

We appreciate your comment and suggestions, we are sure these will highly contribute to the improvement of our manuscript. Here we show a list of the corrections and answers to your question. You can find the corrections in the manuscript since we are using track changes.

Comment:  The article entitled "A one health perspective to recognize Fusarium and Neocosmospora as important in clinical practice" by Valeri Sáenz and coworkers is an interesting attempt to advise regarding the use of fungicides in Agriculture or other field and their impact in human, crop, and animals health.

The article is, in general, well organized and well written. However, in addition to its objective, it introduces a question that is currently under an intense discussion in the Fusarium community, as is the use of the name Neocosmospora for some formerly named Fusarium species. Further, in the manuscript, it is not clear what name the authors are using, and this introduces a point of unnecessary distraction for the reader.

I would suggest the use of only the name of Fusarium for all examples used, waiting to see in the near future what is the conclusion reached by the general Fusarium community. As the authors indicated, there are currently several important articles accepted or being evaluated that go in-depth with this question.

Answer: We agree with the reviewer’s assessment. Accordingly, throughout the manuscript, we have removed the term Neocosmospora

I found some other points that could be improved, which are listed below.

Comment:  Lines 84-86, from In India,.. to terbinafine. This sentence has no sense in this form, please improve it.

Answer: As suggested, the sentence is updated in the revised version of the manuscript, please see line 92-96.

Comment:  Line 92, and all the introduction section. The inclusion of some Fusarium (former) species in the genus Neocosmospora is currently under an intense discussion. However, in the article, authors use the new names even when they indicated at the beginning of the article that will use the name Fusaria-like to for those species under study. Please, authors have to try to be consistent with the nomenclature used in the present article, in which, the objective is not to solve the discrepancies in those species names.

Answer: We agree with the reviewer’s assessment. Accordingly, throughout the manuscript, we have removed the term fusaria-like

Comment:  Line 93, head of section 3.1/ Line 108, head of section 4- and following sections.
See the previous comment.

Answer:  We agree with the reviewer’s assessment. Accordingly, throughout the manuscript, we have removed the term Neocosmospora from all head sections.

Comment:  Section 3.1.  Similarly, to what it was explained for airborne Fusariosis, authors could go further in describing the more frequent fungal infections caused by water transmitted Fusarium conidia.

Answer: Thank you for pointing this out. Although we agree that this is an important consideration, we explain the main role of water in clinical settings, unfortunately there is more information for airborne fusariosis.

Comment:  Line 109. At this point, authors use the term Fusaria-like. Please, the use of this term only in specific points of the article introduces a point of distraction for the reader, please be consistent in the entire article regarding the nomenclature used. This article is interesting itself, authors do not need, I insist, to go further in the nomenclature controversy. Take this into account for the entire article.

Answer: We agree with the reviewer’s assessment. Accordingly, throughout the manuscript, we have removed the term fusaria-like

Comment:  Lines 146-147. I suggest the authors to list the names of the NDMs organisms in order regarding their higher-to-lower incidence. Probably the first in this list should be Fusarium spp.?

Answer: Thank you for pointing this out. Reviewer 1 suggested to omit the list, because this information was not essential in the context of the paper. We have removed the list.

Comment:  Line 148. An additional term is used at this point: fusarium-like. Is this the same as Fusaria-like?. Please revise.

We agree with the reviewer’s assessment. Accordingly, throughout the manuscript, we have removed the term fusarium-like

Comment:  Paragraph starting in line 151. The mechanism described for Fusarium infection in onychomycosis is not exclusive of this fungus, it is the usual way to start this infection for all known onychomycosis-causing agents. Please, improve this paragraph.

Answer: We agree with the reviewer’s assessment. Accordingly, we have removed the sentence in line 186.

Comment:  Lines 201-202. Sentence starting with “The PacC/Rim…..ocular surface” is irrelevant in this manuscript´s context. Maybe, authors could move this sentence to the end of this section.

Answer:  We agree with the reviewer’s assessment. Accordingly, we have removed the sentence in line 290.

Comment: Lines 233-235. From “Azole fungicides….” to “broad-spectrum”. This sentence is out of context.

Answer: Thank you for pointing this out. We have added the word and to connect both sentences that explain about fungicides in line 280.

Comment: Section 6.General comment to this section. 
Authors do not really talk about the fusariosis treatment, this section only includes a summary of only a few antifungal compounds, but it is not an exhaustive review of the diverse possible antifungal treatments. Please, improve or change the head of this section.

Answer: We agree with the reviewer’s assessment. Accordingly, we have changed the head of this section to: Fusariosis treatment (line 275)

Comment:  Section 6.1. Please, include some detail regarding the characteristics of amphotericin B and natamycin, i.e. type, targets… Even when both are common antifungal agents, mainly amphotericin B, it will be easier to follow for non-specialized readers.

Answer: We agree with the reviewer’s assessment. Accordingly, we have included the sentence “(inhibits fungal growth by binding to sterols) in line 312. And (binds to ergosterol in the cell membrane) in line 308.

Comment:  Section 6.2. See previous comment, to be applied for Posaconazole at this point. Please revise further examples of antifungal agents to improve their description in the base of these comments.

Answer: We agree with the reviewer’s assessment. Accordingly, we have included the sentence (inhibits the ergosterol production by binding and inhibiting the lanosterol-14alpha-demethylase) in line 322

Reviewer 4 Report

This review is adequate, I have no major comments. Note that Fusaria is plural, thus write Fusarium species etc. Similarly: fusariosis is a disease, thus do not use fusariosis disease.

The One Health concept is indeed particularly appropriate for Fusarium, which are typical opportunists (not pathogens) in the environment making advantage of any weakness of living organisms. The paper also shows that the taxonomic change to Neocosmospora is unfortunate and confusing: the entire group behaves similarly, it is an ecological (and evolutionary) entity. The use of two names unfortunate.

Author Response

We appreciate your comment and suggestions, we are sure these will highly contribute to the improvement of our manuscript. Here we show a list of the corrections and answers to your question. You can find the corrections in the manuscript since we are using track changes.

Comment:  This review is adequate; I have no major comments. Note that Fusaria is plural, thus write Fusarium species etc. Similarly: fusariosis is a disease, thus do not use fusariosis disease.

The One Health concept is indeed particularly appropriate for Fusarium, which are typical opportunists (not pathogens) in the environment making advantage of any weakness of living organisms. The paper also shows that the taxonomic change to Neocosmospora is unfortunate and confusing: the entire group behaves similarly; it is an ecological (and evolutionary) entity. The use of two names unfortunate.

Answer:  Thank you for your feedback. We agree with the reviewer’s assessment. Accordingly, throughout the manuscript, we have change fusaria to fusarium species and removed the term Neocosmospora and fusarium-like (line 182, 368, 390).

Round 2

Reviewer 1 Report

No the paper is much clearer.

Author Response

Dear Reviewer

We appreciate your careful revision, comments and your very good feedback to our work.   Kind regards,